# UGAE: A Novel Approach to Non-exponential Discounting

## Abstract

The discounting mechanism in Reinforcement Learning determines the relative importance of future and present rewards. While exponential discounting is widely used in practice, non-exponential discounting methods that align with human behavior are often desirable for creating human-like agents. However, non-exponential discounting methods cannot be directly applied in modern on-policy actor-critic algorithms like PPO. To address this issue, we propose Universal Generalized Advantage Estimation (UGAE), which allows for the computation of GAE advantages with arbitrary discounting. Additionally, we introduce Beta-weighted discounting, a continuous interpolation between exponential and hyperbolic discounting, to increase flexibility in choosing a discounting method. To showcase the utility of UGAE, we provide an analysis of the properties of various discounting methods. We also show experimentally that agents with non-exponential discounting trained via UGAE outperform variants trained with Monte Carlo advantage estimation. Through analysis of various discounting methods and experiments, we demonstrate the superior performance of UGAE with Beta-weighted discounting over the Monte Carlo baseline on standard RL benchmarks. UGAE is simple and easily integrated into any advantage-based algorithm as a replacement for the standard recursive GAE.

## 1 Introduction

Building a Reinforcement Learning (RL) algorithm for time-dependent problems requires specifying a discounting mechanism that defines how important the future is relative to the present. Typically, this is done by setting a discount rate $\gamma \in [0, 1]$ and decaying the future rewards exponentially by a factor $\gamma^t$ (Bellman, 1957). This induces a characteristic planning horizon, which should match the properties of the environment. In practice, $\gamma$ is one of the most important hyper-parameters to tune. Small deviations may massively decrease the algorithm's performance, which makes training an RL agent less robust, and more difficult for users to perform a hyper-parameter search with new algorithms and environments.

The choice of a discounting is an example of the bias-variance trade-off. If the discount factor is too low (or correspondingly, the general discounting decreases too rapidly), the value estimate is too biased, and in an extreme case, the agent cannot plan sufficiently far into the future. Conversely, with a discount factor too high, the variance of the value estimation is very large due to the high impact of the distant future, often irrelevant to the decision at hand.

In the literature, a variety of discounting mechanisms have been proposed, from the widely used exponential discounting (Strotz, 1955) to hyperbolic discounting, first introduced by psychologists to describe human behavior (Ainslie & Haslam, 1992), which we show to be two special cases of our Beta-weighted discounting. Other options include fixed-horizon discounting (Lattimore & Hutter, 2011), where all rewards beyond a certain horizon are ignored, or not using any discounting (Naik et al., 2019). Any discounting method can also be truncated by setting it to zero for all timesteps after a certain point.

The Generalized Advantage Estimation (Schulman et al., 2018) (GAE) algorithm, which can be seen as an extension of the TD($\lambda$) algorithm, is widely used in training RL agents, but it can only use exponential discounting. This limits the behaviors that we can observe in trained agents with respect to balancing rewards at multiple timescales. To enable using arbitrary discounting methods, we introduce Universal

General Advantage Estimation (UGAE) – a modified formulation of GAE that accepts any arbitrary discount vectors. We also define a novel discounting method, named Beta-weighted discounting, which is obtained by continuously weighing all exponential discount factors according to a Beta distribution. We show that this method captures both exponential and hyperbolic discounting, depending on its parameters.

Moreover, we offer an analysis of several exponential and non-exponential discounting methods and their properties. While these methods (except for Beta-weighted discounting) are not new, they can be used in practical RL experiments thanks to UGAE; therefore, it is worthwhile to understand their differences.

As pointed out by Pitis (2019), exponential discounting with a constant discount factor fails to model all possible preferences that one may have. While our beta-weighted discounting only introduces a time-dependent discount factor and thus does not solve this problem in its entirety, it enables using more complex discounting mechanisms. Furthermore, our UGAE can serve as a step towards a practical implementation of state-action-dependent discounting.

Finally, we experimentally evaluate the performance of UGAE on a set of RL environments, and compare it to the unbiased Monte Carlo advantage estimation method. We show that UGAE can match or surpass the performance of exponential discounting, without a noticeable increase in computation time. Since it can be seamlessly used with existing codebases (usually by replacing one function), it offers a good alternative to the conventional approach, and enables a large range of future empirical research into the properties of non-exponential discounting.

While currently research into non-exponential discounting is largely limited to toy problems and simple tabular algorithms, our UGAE makes it possible to use arbitrary discounting with state-of-the-art algorithms. It can be used to solve a wide range of problems, including ones with continuous observation and action spaces, and multiagent scenarios, by combining it with algorithms like PPO (Schulman et al., 2017).

In summary, our contributions are twofold: we introduce UGAE, a modification of GAE that accepts arbitrary discounting methods, offering greater flexibility in the choice of a discounting; and we introduce a novel discounting method, named Beta-weighted discounting, which is a practical way of using non-exponential discounting.

## 2 Related Work

**Discounted utility** is a concept commonly used in psychology (Ainslie & Haslam, 1992) and economics (Lazaro et al., 2002) to understand how humans and other animals (Hayden, 2016) choose between a small immediate reward or a larger delayed reward. The simplest model is exponential discounting, where future rewards are considered with exponentially decaying importance. Hyperbolic discounting is sometimes proposed as a more empirically accurate alternative (Ainslie & Haslam, 1992), however it is nonsummable, leading to problems in continuing environments. Other works (Hutter, 2006; Lattimore & Hutter, 2011) consider arbitrary discounting matrices that could vary over time. Schultheis et al. (2022) propose a formal method for non-exponential discounting in continuous-time reinforcement learning under the framework of continuous control.

The same mechanism of discounting future rewards is used in Reinforcement Learning to ensure computational stability and proper handling of hazard in the environment (Sozou, 1998; Fedus et al., 2019). It has been also shown to work as a regularizer (Amit et al., 2020) for Temporal Difference methods, especially when the value of the discount factor is low. The choice to discount future rewards has been criticized from a theoretical standpoint (Naik et al., 2019) as it may favor policies suboptimal with respect to the undiscounted reward. The alternative is using the undiscounted reward for episodic tasks, and the average reward for continuing tasks (Siddique et al., 2020). While this idea is interesting, it has not gained wide acceptance in practice, seeing as setting the discount factor to a nontrivial value often improves the performance of RL algorithms.

In contrast to the hyperbolic discounting of Fedus et al. (2019), we propose Beta-weighted discounting that uses a more general weighing distribution which can be reduced to both exponential and hyperbolic discounting. Furthermore, through our proposed UGAE it is applied to stochastic Actor-Critic algorithms

in discrete-time RL problems, as opposed to the Temporal Difference-based algorithms common in the non-exponential discounting literature (Maia, 2009; Alexander & Brown, 2010), and the continuous-time setting of Schultheis et al. (2022).

**Policy gradient** (PG) algorithms are derived from REINFORCE (Williams, 1992). They directly optimize the expected reward by following its gradient with respect to the parameters of the (stochastic) policy. Many modern algorithms use this approach, such as TRPO (Schulman et al., 2015) as PPO (Schulman et al., 2017). In our work, we use PPO for the experiments. Their central idea is the policy gradient understood as the gradient of the agent's expected returns w.r.t. the policy parameters, with which the policy can be optimized using Gradient Ascent or an algorithm derived from it like Adam (Kingma & Ba, 2015). A practical way of computing the gradient, equivalent in expectation, uses the advantages based on returns-to-go obtained by disregarding past rewards and subtracting a value baseline.

**Advantage estimation** is a useful tool for reducing the variance of the policy gradient estimation without changing the expectation, hence without increasing the bias (Sutton et al., 1999; Mnih et al., 2016). Its key idea is that, instead of using the raw returns, it is more valuable to know how much better an action is than expected. This is measured by the Advantage, equal to the difference between the expected value of taking a certain action in a given state, and the expected value of that state following the policy.

Pitis (2019) introduces a formally justified method of performing discounting with the value of the discount factor being dependent on the current state of the environment and the action taken by the agent. This approach makes it possible to model a wide range of human preferences, some of which cannot be modelled as Markov Decision Processes with a constant exponential discount factor. A similar idea is present in Temporal Difference algorithms (Sutton & Barto, 2018), where their (typically constant) parameters can be state and action-dependent.

**Generalized Advantage Estimation (GAE)** (Schulman et al., 2018) computes the Advantage by considering many ways of bootstrapping the value estimation, weighted exponentially, analogously to TD($\lambda$). At the extreme points of its $\lambda$ parameter, it is equivalent to Temporal Difference or Monte Carlo estimation. GAE has become the standard method of computing advantages in Actor-Critic algorithms (Schulman et al., 2017) due to its simple implementation and the performance improvements.

In contrast, we propose a new, vectorized formulation of GAE that allows using arbitrary discounting mechanisms, including our Beta-weighted discounting, in advantage-based Actor-Critic algorithms. This enables greater flexibility in choosing the discounting method and gives a practical way of doing non-exponential discounting by setting the parameters of Beta-weighted discounting.

## 3    UGAE – Universal Generalized Advantage Estimation

In this section, we introduce the main contribution of this paper, UGAE, which is a way of combining the GAE algorithm (Schulman et al., 2018) with non-exponential discounting methods. Then, we define several discounting methods that will be further explored in this work.

**Problem Setting** We formulate the RL problem as a Markov Decision Process (MDP) (Bellman, 1957). An MDP is defined as a tuple $\mathcal{M} = (\mathcal{S}, \mathcal{A}, P, R, \mu)$, where $\mathcal{S}$ is the set of states, $\mathcal{A}$ is the set of actions, $P: S \times A \to \Delta S$ is the transition function, $R: S \times A \to \mathbb{R}$ is the reward function and $\mu \in \Delta S$ is the initial state distribution. Note that $\Delta X$ represents the set of probability distributions on a given set $X$. An agent is characterized by a stochastic policy $\pi: \mathcal{S} \to \Delta \mathcal{A}$, at each step $t$ sampling an action $a_t \sim \pi(s_t)$, observing the environment's new state $s_{t+1} \sim P(s_t, a_t)$, and receiving a reward $r_t = R(s_t, a_t)$. Over time, the agent collects a trajectory $\tau = \langle s_0, a_0, r_0, s_1, a_1, r_1, \dots \rangle$, which may be finite (episodic tasks) or infinite (continuing tasks).

The typical goal of an RL agent is maximizing the total reward $\sum_{t=0}^{T} r_t$, where $T$ is the duration of the episode (potentially infinite). A commonly used direct objective for the agent to optimize is the total discounted reward $\sum_{t=0}^{T} \gamma^t r_t$ under a given discount factor $\gamma$ (Sutton & Barto, 2018). Using a discount factor can serve as a regularizer for the agent (Amit et al., 2020), and is needed for continuing tasks ($T = \infty$) to ensure that the total reward remains finite.

In this work, we consider a more general scenario that allows non-exponential discounting mechanisms defined by a function $\Gamma^{(\cdot)}\colon \mathbb{N} \to [0,1]$. The optimization objective is then expressed as $R^{\Gamma} = \sum_{t=0}^{\infty} \Gamma^{(t)} r_t$.

### 3.1 UGAE

The original derivation of GAE relies on the assumption that the rewards are discounted exponentially. While the main idea remains valid, the transformations that follow and the resulting implementation cannot be used with a different discounting scheme.

Recall that GAE considers multiple k-step advantages, each defined as:

$$\hat{A}_t^{(k)} = -V(s_t) + \sum_{l=0}^{k-1} \gamma^l r_{t+l} + \gamma^k V(s_{t+k}). \tag{1}$$

Given a weighing parameter $\lambda$, the GAE advantage is then:

$$\hat{A}_t^{GAE(\gamma,\lambda)} := (1-\lambda)(\hat{A}_t^{(1)} + \lambda \hat{A}_t^{(2)} + \dots) = \sum_{l=0}^{\infty} (\gamma\lambda)^l \delta_{t+l}^V$$

where $\delta_t^V = r_t + \gamma V(s_{t+1}) - V(s_t)$. While this formulation makes it possible to compute all the advantages in a dataset with an efficient, single-pass algorithm, it cannot be used with a general discounting method. In particular, $\delta_t^V$ cannot be used as its value depends on which timestep's advantage we are computing.

To tackle this, we propose an alternative expression using an arbitrary discount vector $\Gamma^{(t)}$. To this end, we redefine the k-step advantage using this concept, as a replacement for Equation 1:

$$\tilde{A}_t^{(k)} = -V(s_t) + \sum_{l=0}^{k-1} \Gamma^{(l)} r_{t+l} + \Gamma^{(k)} V(s_{t+k}) \tag{2}$$

We then expand it to obtain the equivalent to Equation 2:

$$\tilde{A}_t^{UGAE(\Gamma,\lambda)} := (1-\lambda)(\tilde{A}_t^{(1)} + \lambda \tilde{A}_t^{(2)} + \dots) \tag{3}$$
$$= -V(s_t) + \sum_{l=0}^{\infty} \lambda^l \Gamma^{(l)} r_{t+l} + (1-\lambda)\sum_{l=0}^{\infty} \Gamma^{(l+1)} \lambda^l V(s_{t+l+1})$$

Note that the second and third terms are both sums of products, and can therefore be interpreted as scalar products of appropriate vectors. By defining $\boldsymbol{r}_t = [r_{t+i}]_{i\in\mathbb{N}}$, $\boldsymbol{V}_t = [V(s_{t+i})]_{i\in\mathbb{N}}$, $\boldsymbol{\Gamma} = [\Gamma^{(i)}]_{i\in\mathbb{N}}$, $\boldsymbol{\Gamma}' = [\Gamma^{(i+1)}]_{i\in\mathbb{N}}$, $\boldsymbol{\lambda} = [\lambda^i]_{i\in\mathbb{N}}$, we rewrite Equation 3 in a vectorized form in Equation 4. We use the notation that $\boldsymbol{x} \odot \boldsymbol{y}$ represents the Hadamard (element-wise) product, and $\boldsymbol{x} \cdot \boldsymbol{y}$ – the scalar product.

**Theorem 1.** *UGAE: GAE with arbitrary discounting*

*Consider $\boldsymbol{r}_t, \boldsymbol{V}_t, \boldsymbol{\Gamma}, \boldsymbol{\Gamma}', \boldsymbol{\lambda}, \lambda$ defined as above. We can compute GAE with arbitrary discounting as:*

$$\tilde{A}_t^{UGAE(\Gamma,\lambda)} := -V(s_t) + (\boldsymbol{\lambda} \odot \boldsymbol{\Gamma}) \cdot \boldsymbol{r}_t + (1-\lambda)(\boldsymbol{\lambda} \odot \boldsymbol{\Gamma}') \cdot \boldsymbol{V}_{t+1}$$

*If $\Gamma^{(t)} = \gamma^t$, this is equivalent to the standard GAE advantage.* **Proof** *is in the supplemental material.*

**Discussion.** Theorem 1 gives a vectorized formulation of GAE. This makes it possible to use GAE with arbitrary discounting methods with little computational overhead, by leveraging optimized vector computations.

Note that while the complexity of exponential GAE computation for an entire episode is $O(T)$ where $T$ is the episode length, the vectorized formulation increases it to $O(T^2)$ due to the need for multiplying large vectors. Fortunately, truncating the future rewards is trivial using the vectorized formulation, and that can be used

through truncating the discounting horizon, by setting a maximum length $L$ of the vectors in Theorem 1. The complexity in that case is $O(LT)$, so again linear in $T$ as long as $L$ stays constant. In practice, as we show in this paper, the computational cost is not of significant concern, and the truncation is not necessary, as the asymptotic complexity only becomes noticeable with unusually long episodes.

## 3.2 Added estimation bias

An important aspect of our method is the additional bias it introduces to the value estimation. To compute a k-step advantage, we must evaluate the tail of the reward sequence using the discounting itself (the $V(s_{t+k})$ term in Equation 1). This is impossible with any non-exponential discounting, as the property $\Gamma^{(k+t)} = \Gamma^{(k)}\Gamma^{(t)}$ implies $\Gamma^{(\cdot)}$ being an exponential function. Seeing as we are performing an estimation of the value of those last steps, this results in an increase in the estimation bias compared to Monte Carlo estimation.

This ties into the general bias-variance trade-off when using GAE or TD-lambda estimation. In its original form, it performs interpolation between high-variance (Monte Carlo) and high-bias (TD) estimates for exponential discounting. In the case of non-exponential discounting, using UGAE as opposed to Monte Carlo estimates has the same effect of an increase in bias, but decreasing the variance in return.

The difference between the non-exponential discounting and its tail contributes to an additional increase of bias beyond that caused by using GAE, but we show that this bias remains finite in the infinite time horizon for summable discountings (including our Beta-weighted discounting, as well as any truncated discounting).

**Theorem 2.** *UGAE added bias*

*Consider an arbitrary summable discounting $\Gamma^{(t)}$. The additional bias, defined as the discrepancy between the UGAE and Monte Carlo value estimations, is finite in the infinite time horizon.* **Proof** *is in the supplemental material.*

In practice, as we show in our experiments, the decreased variance enabled by UGAE effectively counteracts the added bias, resulting in an overall better performance over Monte Carlo estimation.

## 4 Beta-weighted discounting

In this section, we present our second contribution, i.e. Beta-weighted discounting. It uses the Beta distribution as weights for all values of $\gamma \in [0, 1]$. We use their expected value as the effective discount factors $\Gamma^{(t)}$, which we show to be equal to the distribution's raw moments.

### 4.1 Beta-weighted discounting

As a simple illustrative example, let us use two discount factors $\gamma_1$, $\gamma_2$ with weights $p$ and $(1-p)$ respectively, where $p \in [0, 1]$. This can be treated as a multiple-reward problem (Shelton, 2000) where the total reward is a weighted sum of individual rewards $R^\gamma = \sum_{t=0}^{\infty} \gamma^t r_t$. Therefore, we have:

$$R^{(\gamma_1, \gamma_2)} = p \sum \gamma_1^t r_t + (1-p) \sum \gamma_2^t r_t \tag{4}$$
$$= \sum r_t (p\gamma_1^t + (1-p)\gamma_2^t)$$

We extend this reasoning to any countable number of exponential discount factors with arbitrary weights, that sum up to 1. Taking this to the continuous limit, we also consider continuous distributions of discount factors $w \in \Delta([0, 1])$, leading to the equation:

$$R^w = \sum r_t \int_0^1 w(\gamma)\gamma^t dt = \sum r_t \Gamma^{(t)} \tag{5}$$

An important observation is that as long as $\text{supp}(w) \subseteq [0, 1]$, the integral is by definition equal to the t-th raw moment of the distribution $w$. Hence, with an appropriately chosen distribution. an analytical expression is obtained for all its moments, and therefore, all the individual discount factors $\Gamma^{(t)}$.

We choose the Beta distribution due to the simple analytical formula for its moments, as well as its relation to other common discounting methods. Its probability density function is defined as $f(x; \alpha, \beta) \propto x^{\alpha-1}(1-x)^{\beta-1}$. Note that with $\beta = 1$, it is equivalent to the exponential distribution which induces a hyperbolic discounting. Its moments are known in analytical form (Johnson et al., 1994), which leads to our proposed discounting mechanism in Theorem 3.

**Theorem 3.** *Beta-weighted discounting*

*Consider $\alpha, \beta \in [0, \infty)$. The following equations hold for the Beta-weighted discount vector parametrized by $\alpha, \beta$.* **Proof** *is in the supplementary material.*

$$\Gamma^{(t)} = \prod_{k=0}^{t-1} \frac{\alpha + k}{\alpha + \beta + k} \tag{6}$$

$$\Gamma^{(t+1)} = \frac{\alpha + t}{\alpha + \beta + t} \Gamma^{(t)} \tag{7}$$

## 4.2 Beta distribution properties

Here, we investigate the Beta distribution's parameter space and consider an alternative parametrization that eases its tuning. We also analyze important properties of the Beta-weighted discounting based on those parameters, and compare them to exponential and hyperbolic baselines.

Canonically, the Beta distribution is defined by parameters $\alpha, \beta \in (0, \infty)$. It is worth noting certain special cases and how this approach generalizes other discounting methods. When $\alpha, \beta \to \infty$ such that its mean $\mu := \frac{\alpha}{\alpha+\beta} = const.$, the beta distribution asymptotically approaches the Dirac delta distribution $\delta(x - \mu)$, resulting in the usual exponential discounting $\Gamma^{(t)} = \mu^t$. Alternatively, when $\beta = 1$, we get $\Gamma^{(t)} = \prod_{k=0}^{t-1} \frac{\alpha+k}{\alpha+k+1} = \frac{\alpha}{\alpha+t} = \frac{1}{1+t/\alpha}$, i.e. hyperbolic discounting.

**Mean $\mu$ and dispersion $\eta$** A key property is that we would like the effective discount rate to be comparable with existing exponential discount factors. To do so, we define a more intuitive parameter to directly control the distribution's mean as $\mu = \frac{\alpha}{\alpha+\beta} \in (0, 1)$. $\mu$ defines the center of the distribution and should therefore be close to typically used $\gamma$ values in exponential discounting.

A second intuitive parameter should control the dispersion of the distribution. Depending on the context, two choices seem natural: $\beta$ itself, or its inverse $\eta = \frac{1}{\beta}$. As stated earlier, $\beta$ can take any positive real value. By discarding values of $\beta < 1$ which correspond to a local maximum of the probability density function around 0, we obtain $\eta \in (0, 1]$. That way we obtain an easy-to-interpret discounting strategy. as we show in Lemma 4, $\eta \to 0$ and $\eta = 1$ correspond to exponential discounting, and hyperbolic discounting, respectively, which allows us to finally define the range of $\eta$ as $[0, 1]$. Other values smoothly interpolate between both of these methods, similar to how GAE interpolates between Monte Carlo and Temporal Difference estimation.

Given the values of $\mu$ and $\eta$, the original distribution parameters can be recovered as $\alpha = \frac{\mu}{\eta(1-\mu)}$ and $\beta = \frac{1}{\eta}$. The raw moments parametrized by $\mu$ and $\eta$ are $m_t = \prod_{k=0}^{t-1} \frac{\mu + k\eta(1-\mu)}{1 + k\eta(1-\mu)}$.

**Lemma 4.** *Special cases of Beta-weighted discounting*

*We explore the relation of Beta-weighted discounting to exponential and hyperbolic discountings. Consider the Beta-weighted discounting $\Gamma^{(t)}$ parametrized by $\mu \in (0, 1), \eta \in (0, 1]$. The following is true:*

- *if $\eta \to 0$, then $\Gamma^{(t)} = \mu^t$, i.e. it is equal to exponential discounting*
- *if $\eta = 1$, then $\Gamma^{(t)} = \frac{\mu}{\mu+(1-\mu)t} = \frac{1}{1+t/\alpha}$, i.e. it is equal to hyperbolic discounting*

**Proof** *is in the supplemental material.*

**Discussion.** Beta-weighted discounting is controlled by two parameters $(u, \eta)$, which includes the classic exponential discounting, but also enables more flexibility in designing agent behaviors .

**Lemma 5.** *Beta-weighted discounting summability*

*Given the Beta-weighted discount vector $\Gamma^{(t)} = \prod_{k=0}^{t-1} \frac{\alpha+k}{\alpha+\beta+k}$, $\alpha \in [0,\infty)$, $\beta \in [0,\infty)$, the following property holds.* **Proof** *is in the supplemental material.*

$$\sum_{t=0}^{\infty} \Gamma^{(t)} = \begin{cases} \frac{\alpha+\beta-1}{\beta-1} & \text{if } \beta > 1 \\ \infty & \text{otherwise} \end{cases} \tag{8}$$

**Discussion.** Lemma 5 describes the conditions under which the Beta-weighted discounting is summable depending on its parameters. While less critical for episodic tasks, summability of the discount function is important for continuing tasks. Otherwise, the discounted reward can grow arbitrarily high over time.

## 5 Analysis of non-exponential discounting methods

Here, our goal is to justify the usage, and enable deep understanding of different discounting methods. To this end, we first analyze some of their main properties: the importance of future rewards, the variance of the discounted rewards, the effective planning horizon, and the total sum of the discounting. Then, we compare those properties among the previously described discounting methods.

Since not all discounting methods are summable (particularly the cases of hyperbolic and no discounting), we consider the maximum ("infinite") episode length to be 10000 steps. We focus on a characteristic time scale of the environment around 100 steps.

### 5.1 Properties of discounting

**Importance of future rewards** Properly describing the influence of the future under a specific discounting is challenging. On one hand, individual rewards are typically counted with a smaller weight, as discount vectors $\Gamma^{(t)}$ are usually monotonically decreasing. On the other hand, the longer the considered time horizon is, the more timesteps it includes, increasing its overall importance. Furthermore, a long time horizon (e.g. 100 steps) directly includes a shorter horizon (e.g. 10 steps), and therefore, the partial sums are not directly comparable. To balance these aspects, we focus on the importance of the first 100 steps using the following expressions:

$$\Gamma_{t_1}^{t_2} = \frac{\sum_{t=t_1}^{t_2} \Gamma^{(t)}}{\sum_{t=0}^{\infty} \Gamma^{(t)}} \tag{9}$$

**Variance of the discounted rewards** The overall objective of the RL agent is maximizing the total (discounted) reward it obtains in the environment. Since both the policy and the environment can be stochastic, the total reward will also be subject to some uncertainty. While the exact rewards, as well as their variances, depend heavily on the exact environment, we make the simplifying assumption that the instantaneous rewards $r_t$ are sampled according to a distribution $D$ with a constant variance of $\sigma^2$, e.g. $r_t \sim D = \mathcal{N}(\mu, \sigma^2)$ with an arbitrary, possibly varying, $\mu$. We also assume all rewards to be uncorrelated, which leads to the following expression:

$$\text{Var}[\sum_{t=0}^{T} \Gamma^{(t)} r_t] = \sum_{t=0}^{T} \Gamma^{(t)^2} \text{Var}[r_t] = \sum_{t=0}^{T} \Gamma^{(t)^2} \sigma^2 = \sigma^2 \sum_{t=0}^{T} \Gamma^{(t)^2}$$

Equation 10 shows that the variance of the total discounted reward is proportional to the sum of all the squares of discount factors. While in some cases it is easy to obtain analytically, quite often the expression can be complex and difficult to obtain and analyze; hence, we consider the numerical values, as well as analytical expressions where applicable.

**Effective planning horizon** For any discounting $\Gamma^{(t)}$, our goal is to have a measure of its effective planning horizon. However, in most cases, we cannot have a clear point beyond which the rewards do not matter and there is not a unique notion of a time horizon that could be specified. Thus, to maintain consistency with

the standard notion of a time horizon from exponential discounting, we define the effective time horizon as the timestep $T_{eff}$ after which approximately $\frac{1}{e}(\approx 37\%)$ of the weight of the discounting remains.

**Total sum of the discounting** Depending on the RL algorithm and the reward normalization method (if any), the magnitude of the total discounted reward might impact the stability of the training, as neural networks typically cannot deal with very large numbers. For this reason, we consider the sum of all rewards under a given discounting.

**Consistency** It is worth keeping in mind that, as pointed out by Lattimore & Hutter (2011), the only time consistent discounting is the exponential discounting. This means that it is possible that other methods cause the agent to change its plans over time. While this has the potential to significantly degrade the performance in some specific environments, that is often not the case in typical RL tasks, as we show in Section 6.

## 5.2 Discounting methods

**Beta-weighted Discounting** is described in the following section. Exponential and hyperbolic discountings are equivalent to Beta-weighted discounting with $\eta{=}0$ and $\eta{=}1$, respectively. The former is given by $\Gamma^{(t)}{=}\mu^t$ with $\mu{\in}[0,1]$, and the latter by $\Gamma^{(t)}{=}\frac{1}{1+kt}{=}\frac{1}{1+\frac{1-\mu}{\mu}t}$ parametrized by $k{\in}[0,\infty)$ or $\mu{\in}(0,1]$.

**No Discounting** implies that the discount vector takes the form $\forall_{t\in\mathbb{N}}\Gamma^{(t)}{=}1$. This is nonsummable, but it is trivial to compute any partial sums to estimate the importance of future rewards or the variance. The effective planning horizon depends on the episode length $T$, and is equal to $(1 - \frac{1}{e})T$.

**Fixed-Horizon Discounting** Here, there is a single parameter $T_{max}$ which defines how many future rewards are considered with a constant weight. The discount vector is then $\Gamma^{(t)} = \mathbf{1}_{t<T_{max}}$, and the planning horizon, according to our definition, is $(1 - \frac{1}{e})T_{max}$.

**Truncated Discounting** All discounting methods can be truncated by adding an additional parameter $T_{max}$, and setting the discount vector to 0 for all timesteps $t > T_{max}$. Truncating a discounting decreases the importance of future rewards and the effective planning horizon, but also decreases the variance of the total rewards.

## 5.3 Experimental Analysis

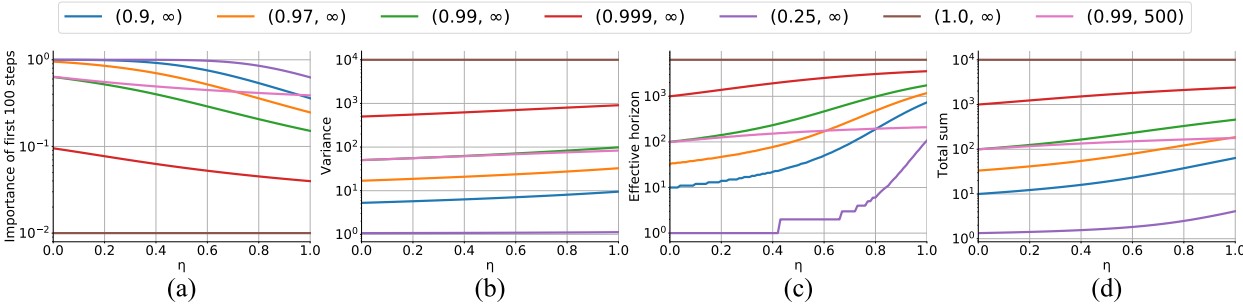

Figure 1: Different properties of a discounting, as a function of $\eta$, with given $(\mu, T_{max})$ parameters listed in the legend. (a) Importance of the near future (b) Variance measure (c) Effective time horizon (d) Total discounting sum

To analyze the properties of different discounting methods, we compute them for a set of relevant discounting methods and their parameters. The impact of Beta discounting's $\eta$ is illustrated in Figure 1(a-d), showing: (a) the importance of first 100 steps $\Gamma_0^{100}$, (b) the variance measure, (c) the effective time horizon, (d) the total sum of the discounting – full results are in the supplement. Note that the choice of the discounting is an example of the common bias-variance trade-off. If the real objective is maximizing the total undiscounted

reward, decreasing the weights of future rewards inevitably biases the algorithm's value estimation, while simultaneously decreasing its variance.

Using **no discounting**, the rewards from the distant future have a dominant contribution to the total reward estimate since more steps are included. **Exponential discounting** places more emphasis on the short term rewards, according to its $\gamma$ parameter, while simultaneously decreasing the variance; when $\gamma = 0.99$, it effectively disregards the distant future of $t > 1000$.

In **Beta-weighted discounting** with $\eta > 0$, the future rewards importance, the variance and the effective time horizon increase with $\eta$. If $\mu$ is adjusted to make $T_{eff}$ similar to exponential discounting's value, the variance decreases significantly, and the balance of different time horizons shifts towards the future, maintaining some weight on the distant future.

With **hyperbolic discounting** (Beta-weighted with $\eta = 1$) the distant future becomes very important, and the effective time horizon becomes very large (in fact, infinite in the limit of an infinitely long episode). To reduce the time horizon to a value close to 100, its $\mu$ parameter has to be very small, near $\mu = 0.25$, putting most of the weight on rewards that are close in time, but also including the distant rewards unlike any exponential discounting.

The behavior of **fixed-horizon discounting** is simple – it acts like no discounting, but ignores any rewards beyond $T_{max}$. Truncating another discounting method results in ignoring the rewards beyond $T_{max}$, and decreasing the variance and the effective time horizon (see supplemental material).

In summary, by modifying $\eta$ in Beta-weighted discounting, and $T_{max}$ in the truncated scenario, we can successfully change various aspects of the resulting discounting in a more flexible way than with exponential discounting. In general, changing the discounting method can bias the agent to favor certain timescales and focus on maximizing the rewards that occur within them.

### 5.4 Discussion

When $\eta$ is increased, the importance shifts towards the future (Figure 1a), variance (Figure 1b) and the effective horizon (Figure 1c) increase. Introducing a truncation (pink line) decreases the effective horizon and shifts the reward importance towards the near rewards. When truncated at 100 steps, Beta-weighted discounting with $\eta = 0.5$ and hyperbolic discounting ($\eta = 1$) have very similar properties, indicating their main difference lies in how they deal with the distant future. This is confirmed by comparing the non-truncated versions, where hyperbolic discounting puts a very large weight on the distant future, unlike the Beta-weighted discounting.

### 5.5 Why non-exponential discounting?

A natural question that arises during this discussion is – why do we even want to train agents with non-exponential discounting? As described by Naik et al. (2019), optimizing a discounted reward is not equivalent to optimizing the reward itself. The choice of a discounting method affects the optimal policy, or even its existence in the first place. While we do not tackle this problem in this work, we enable larger flexibility in designing the discounting mechanism. This in turn allows researchers to generate more diverse emergent behaviors through the choice of an appropriate discounting – in case that exponential discounting leads to undesirable results.

As mentioned earlier, any discounting other than exponential has the potential for inconsistent behavior. This means that the agent may change its mind on a decision as time passes, without any additional changes to the situation. While this behavior is admittedly irrational, it is commonly exhibited by humans (Ainslie & Haslam, 1992). Therefore, it is important to take this into consideration when creating agents meant to mimic human behavior in applications like video games or social robotics, where human-likeness is important, and potentially not appropriately reflected in the reward function.

## 6 DRL Experiments

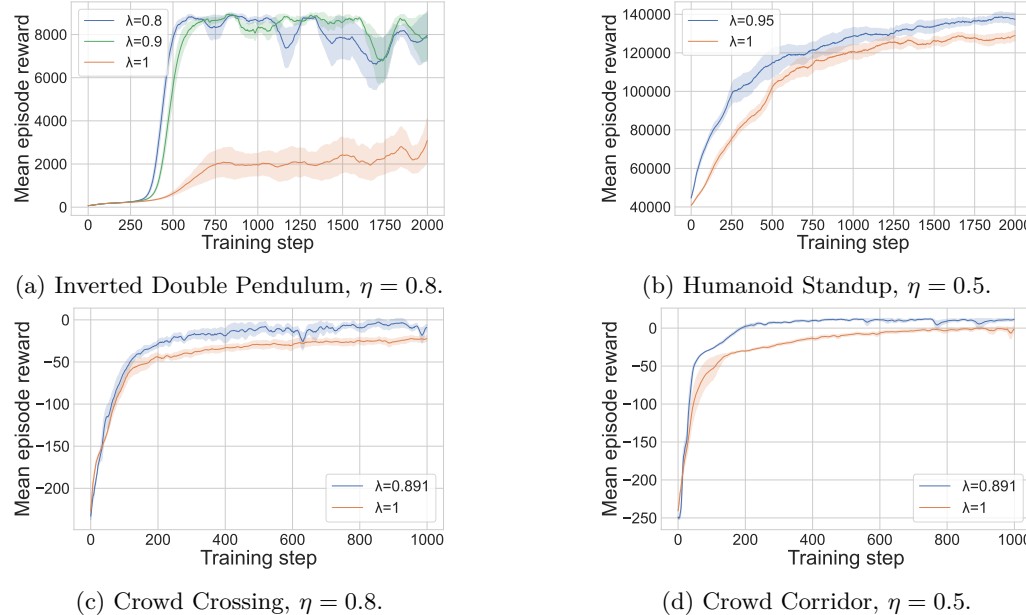

(a) Inverted Double Pendulum, $\eta = 0.8$.      (b) Humanoid Standup, $\eta = 0.5$.

(c) Crowd Crossing, $\eta = 0.8$.      (d) Crowd Corridor, $\eta = 0.5$.

Figure 3: Training curves in DRL experiments using non-exponential discounting. All curves are averaged across 8 independent training runs. Shading indicates the standard error of the mean. In all experiments, using $\lambda$ values that were tuned for optimality with exponential discounting, UGAE significantly outperforms the MC baseline ($\lambda = 1$). This indicates that UGAE enables translating the benefits of GAE to non-exponential discounting.

In this section, we evaluate our UGAE for training DRL agents with non-exponential discounting, in both single-agent and multiagent environments. As the baseline, we use non-exponential discounting with regular Monte Carlo (MC) advantage estimation, equivalent to UGAE with $\lambda = 1$. We use Beta-weighted discounting to parametrize the non-exponential discounting with its $\eta$ value. The code with hyperparameters and other implementation details will be released upon publication.

We use four test environments to evaluate our discounting method: InvertedDoublePendulum-v4 and HumanoidStandup-v4 from MuJoCo via Gym (Todorov et al., 2012; Brockman et al., 2016); Crossway and Corridor crowd simulation scenarios with 50 homogeneous agents each, introduced by Kwiatkowski et al. (2023). The crowd scenarios are

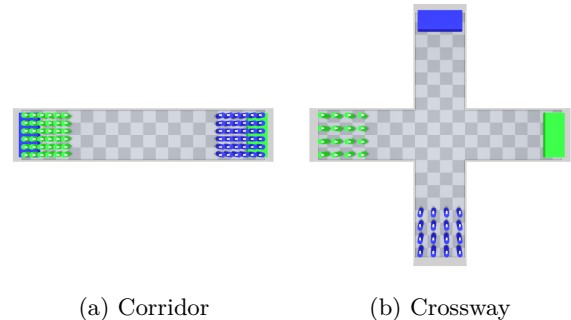

(a) Corridor      (b) Crossway

Figure 2: Visualizations of the two crowd simulation scenarios used in the experiments. In both cases, each agent needs to reach the opposite end of their respective route, and is then removed from the simulation.

displayed in Figure 2. We chose these environments because their optimal $\lambda$ values with exponential GAE are relatively far from $\lambda = 1$ based on prior work. In environments where GAE does not provide benefit over MC estimation, we similarly do not expect an improvement with non-exponential discounting.

Inverted Double Pendulum and Humanoid Standup are both skeletal control tasks with low- and high-dimensional controls, respectively. The former has an 11-dimensional observation space and 1-dimensional action space, whereas the latter has a 376-dimensional observation space, and a 17-dimensional action space. The crowd simulation scenarios use a hybrid perception model combining raycasting to perceive walls, and direct agent perception for neighboring agents for a total of 177-dimensional vector observation and 4-dimensional embedding of each neighbor, as described in Kwiatkowski et al. (2023). They use a 2-dimensional

action space with polar velocity dynamics. The episode length is 1000 for MuJoCo experiments, and 200 for crowd simulation experiments.

We train the agents with the PPO algorithm, with hyperparameters based on the RL Baselines Zoo (Raffin, 2020) for the MuJoCo environments, and from Kwiatkowski et al. (2023) for the crowd environments. It is worth noting that the MuJoCo hyperparameters have been tuned for a prior version of the environments (v3), and thus the results can be different. We use the optimal value of $\lambda$ in the exponential discounting paradigm, and apply it analogously with UGAE. A single training takes 2-5 hours with a consumer GPU.

### 6.1 Results

We show the results in Figure 3. In all tested environments, training with UGAE leads to a higher performance compared to the MC baseline, with the largest effect being present in the Inverted Double Pendulum where UGAE achieves a mean episode reward of **8213 $\pm$ 1067**, while MC only achieves **3364 $\pm$ 1078**. The effect is smaller in the Humanoid Standup task, but still significant, with the final rewards being **137300 $\pm$ 3400** and **129000 $\pm$ 2520** respectively. In the crowd scenarios, a more detailed analysis of the emergent behaviors indicates that agents trained with MC fail to maintain a comfortable speed (which is part of the reward function), while UGAE agents are able to efficiently navigate to their goals. This results in rewards of **11.2 $\pm$ 1.457** for UGAE and **-0.156 $\pm$ 2.745** for MC in the Corridor scenario; and **-1.46 $\pm$ 1.50** for UGAE and **-35.15 $\pm$ 8.81** for MC in the Crossway scenario.

### 6.2 Computation time

To estimate the computational impact of our vectorized UGAE formulation as compared to the standard recursive GAE, we generate 16 random episodes with a length between 1 and 100,000 steps, and plot the time needed to compute the advantages as a function of the episode length. The results are in Figure 4. For a reference duration of a full training step, we use 10 seconds. It shows that while the computational cost of UGAE (blue) is larger than that of GAE (orange line), it remains insignificant compared to a full training step with episodes shorter than $10^4$ steps. For longer episodes, it becomes noticeable, however, this is rarely the case in practice.

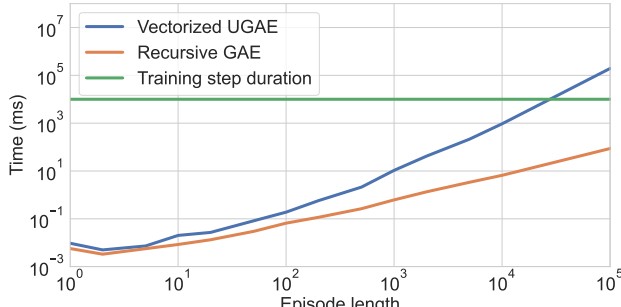

Figure 4: Time needed to compute GAE (orange) and UGAE (blue) with a single consumer CPU, on log-log scale. The green line is a reference duration of 10 seconds representing a typical training iteration. While UGAE is more expensive, with typical training step durations, the time to compute its values is negligible.

### 6.3 Discussion

As our experiments show, using UGAE with episodes of length up to ca. $10^3$ steps carries a negligible computational cost, allowing its seamless integration into a PPO training pipeline. At the same time, it enables a performance improvement mirroring that of GAE, but for non-exponential discounting. In conjunction with Beta-weighted discounting, it enables practical and efficient training of agents with non-exponential discounting.

The main limitation of our work lies in the asymptotic complexity of advantage computation. The time needed to compute the UGAE advantage is negligible with episodes up to around $10^3$ steps, and becomes noticeable (although still not overwhelmingly so) at around $10^4$ steps. In the rare scenario one needs to compute the advantages for episodes over $10^4$ steps, the computation may become too expensive and require truncating the discounting, or otherwise optimizing it.

Our beta-weighted discounting adds a new hyperparameter, which is a potential challenge, as RL algorithms typically already have a large number of hyperparameters that must be optimized. However, due to the

interpretation of $\eta$, there is a natural default value of $\eta = 0$ which corresponds to exponential discounting. With the other extreme being $\eta = 1$, this yields a compact range of possible values that can be easily included in a hyperparameter optimization procedure. This also opens the door to further research on automatically tuning the discount factor from a wider family of possibilities as opposed to just exponential methods.

## 7 Conclusions

Our work follows the exciting trend of rethinking the discounting mechanism in RL. In typical applications, our UGAE can be used with negligible overhead, and together with Beta-weighted discounting they provide an elegant way to perform efficient non-exponential discounting. To our knowledge, UGAE is the first method that enables using arbitrary discounting mechanisms in Actor-Critic algorithms. Our experiments show that using non-exponential discounting gives more flexibility in the temporal properties of the RL agent, and thus enables more diverse, potentially human-like, emergent behaviors.

Importantly, this work makes it possible for researchers to empirically investigate different methods of discounting and their relation with various RL problems, including state-dependent discounting. A challenging but valuable contribution would be developing a method to analyze the properties of an environment, and relating them to the ideal discounting method. Finally, developing an analogous method for value-based algorithms like DQN or DDPG would further broaden the applicability of non-exponential discounting across a more comprehensive selection of cutting-edge RL algorithms. In doing so, we anticipate that our work will inspire and inform the ongoing evolution of reinforcement learning and its diverse applications.

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

## Supplemental Material

## A Proofs

**Notation**

- $\Gamma^{(t)}$ – general discount factor at step $t$

- $r_t$ – reward at step $t$

- $V(s)$ – value estimate of a state $s$

- $\Gamma(z) = \int_0^\infty x^{z-1}e^{-x}dx$ – the usual Gamma function

- $B(\alpha, \beta) = \int_0^1 t^{\alpha-1}(1-t)^{\beta-1}dt = \frac{\Gamma(\alpha)\Gamma(\beta)}{\Gamma(\alpha+\beta)}$

- $\Delta(X)$ – set of probability distributions on the set $X$

**Theorem 1.** *UGAE: GAE with arbitrary discounting*

*Consider $\boldsymbol{r}_t = [r_{t+i}]_{i\in\mathbb{N}}$, $\boldsymbol{V}_t = [V(s_{t+i})]_{i\in\mathbb{N}}$, $\boldsymbol{\Gamma} = [\Gamma^{(i)}]_{i\in\mathbb{N}}$, $\boldsymbol{\Gamma}' = [\Gamma^{(i+1)}]_{i\in\mathbb{N}}$, $\boldsymbol{\lambda} = [\lambda^i]_{i\in\mathbb{N}}$. We define the GAE with arbitrary discounting as:*

$$\tilde{A}_t^{UGAE(\Gamma,\lambda)} := -V(s_t) + (\boldsymbol{\lambda}\odot\boldsymbol{\Gamma})\cdot\boldsymbol{r}_t + (1-\lambda)(\boldsymbol{\lambda}\odot\boldsymbol{\Gamma}')\cdot\boldsymbol{V}_{t+1} \tag{10}$$

*If $\Gamma^{(t)} = \gamma^t$, this is equivalent to the standard GAE advantage.*

*Proof.* Recall that we defined the k-step advantage as

$$\tilde{A}_t^{(k)} := -V(s_t) + \sum_{l=0}^{k-1}\Gamma^{(l)}r_{t+l} + \Gamma^{(k)}V(s_{t+k}).$$

With this, we expand the expression for UGAE as

$$\begin{aligned}
\tilde{A}_t^{UGAE(\Gamma,\lambda)} &= (1-\lambda)(\tilde{A}_t^{(1)} + \lambda\tilde{A}_t^{(2)} + \lambda^2\tilde{A}_t^{(3)} + \ldots) \\
&= (1-\lambda)\left[-V(s_t) + r_t + \Gamma^{(1)}V(s_{t+1}) - \lambda V(s_t) + \lambda r_t + \lambda\Gamma^{(1)}r_{t+1} + \lambda\Gamma^{(2)}V(s_{t+2}) + \ldots\right] \\
&= (1-\lambda)\left[-\sum_{l=0}^\infty(\lambda^l)V(s_t) + \sum_{l=0}^\infty(\lambda^l)r_t + \sum_{l=1}^\infty(\lambda^l)\Gamma^{(1)}r_1 + \ldots + \Gamma^{(1)}V(s_{t+1}) + \lambda\Gamma^{(2)}V(s_{t+2}) + \ldots\right] \\
&= (1-\lambda)\left[\frac{-V(s_t)}{1-\lambda} + \frac{r_t}{1-\lambda} + \frac{\lambda\Gamma^{(1)}r_{t+1}}{1-\lambda} + \ldots + \Gamma^{(1)}V(s_{t+1}) + \lambda\Gamma^{(2)}V(s_{t+2}) + \ldots\right] \\
&= -V(s_t) + \sum_{l=0}^\infty\lambda^l\Gamma^{(l)}r_{t+l} + (1-\lambda)\sum_{l=0}^\infty\lambda^l\Gamma^{(l+1)}V(s_{t+l+1}) \\
&= -V(s_t) + (\boldsymbol{\lambda}\odot\boldsymbol{\Gamma})\cdot\boldsymbol{r}_t + (1-\lambda)(\boldsymbol{\lambda}\odot\boldsymbol{\Gamma}')\cdot\boldsymbol{V}_{t+1} \tag{11}
\end{aligned}$$

showing the validity of Equation 5 in the main manuscript. To reduce it to standard GAE with exponential discounting, it is sufficient to replace $\Gamma^{(\cdot)}$ with $\gamma^\cdot$ in the second line of Equation 11 and follow the proof from Schulman et al. (2018).

$\square$

**Theorem 2.** *UGAE added bias*

*Consider an arbitrary summable discounting $\Gamma^{(t)}$ in an environment where the reward is bounded by $R \in \mathbb{R}$. The additional bias, defined as the discrepancy between the UGAE and Monte Carlo value estimations, is finite.*

*Proof.* The goal is to find a finite bound on the difference between the empirical (Monte Carlo) value estimate used for bootstrapping the advantage estimation, and the value estimation used in UGAE, in the infinite time limit. This can be expressed as follows:

$$\left| \sum_{l=0}^{\infty} \Gamma^{(l+1)} \lambda^l \hat{V}(s_{t+l+1}) - \sum_{l=0}^{\infty} \lambda^l V_{l+1}^{\Gamma}(s_{t+l+1}) \right| \tag{12}$$

where $V_{l+1}^{\Gamma}$ is the true value as discounted with $\Gamma$, $l$ steps after the step for which we compute the advantage, and $\hat{V}$ is the UGAE estimate:

$$V_k^{\Gamma}(s_t) = \mathbb{E} \sum_{t'} \Gamma^{(k+t')} r_{t'} \tag{13}$$

$$\hat{V}_k(s_t) = \mathbb{E} \sum_{t'} \Gamma^{(t')} r_{t'} \tag{14}$$

With this we can expand the expression in Equation 12 as follows:

$$\left| \sum_{l=0}^{\infty} \Gamma^{(l+1)} \lambda^l \hat{V}(s_{t+l+1}) - \sum_{l=0}^{\infty} \lambda^l V_{l+1}^{\Gamma}(s_{t+l+1}) \right| =$$

$$= \left| \sum_{l=0}^{\infty} \lambda^l \left[ \Gamma^{(l+1)} \hat{V}(s_{t+l+1}) - V_{l+1}^{\Gamma}(s_{t+l+1}) \right] \right| =$$

$$= \left| \sum_{l=0}^{\infty} \lambda^l \left[ \Gamma^{(l+1)} \mathbb{E} \sum_{t'} \Gamma^{(t')} r_{t'} - \mathbb{E} \sum_{t'} \Gamma^{(l+1+t')} r_{t'} \right] \right| =$$

$$= \mathbb{E} \left| \sum_{l=0}^{\infty} \lambda^l \sum_{t'} \left[ \Gamma^{(l+1)} \Gamma^{(t')} - \Gamma^{(l+1+t')} \right] r_{t'} \right| \leq$$

$$\leq \mathbb{E} \left| \sum_{l=1}^{\infty} \lambda^{(l-1)} \sum_{t'} \left[ \Gamma^{(l)} \Gamma^{(t')} - \Gamma^{(l+t')} \right] \right| R \tag{15}$$

We now focus on the key expression of the last line, which we denote as $\delta_l^{\Gamma}$:

$$\delta_l^{\Gamma} = \sum_{t'} \Gamma^{(l)} \Gamma^{(t')} - \Gamma^{(l+t')} \leq$$

$$\leq \sum_{t'} \Gamma^{(l)} \Gamma^{(t')} \leq$$

$$\leq \max_t \Gamma^{(t)} \sum_{t'} \Gamma^{(t')} \leq \infty \tag{16}$$

This shows that $\delta_l^{\Gamma}$ is finite for any summable discounting $\Gamma$ and for every value of $l$. Because the $\delta_l^{\Gamma}$ terms are summed with an exponentially decreasing factor $\lambda^{(l-1)}$ in Equation 15, the total difference in Equation 12 must also be finite, completing the proof.

$\square$

**Theorem 3.** *Beta-weighted discounting*

*Consider $\alpha, \beta \in [0, \infty)$. The following equations hold for the Beta-weighted discount vector parametrized by $\alpha, \beta$:*

$$\Gamma^{(t)} = \prod_{k=0}^{t-1} \frac{\alpha + k}{\alpha + \beta + k} \tag{17}$$

$$\Gamma^{(t+1)} = \frac{\alpha + t}{\alpha + \beta + t} \Gamma^{(t)} \tag{18}$$

*Proof.* As mentioned in the paper, if we use an effective discount factor obtained by weighing individual values according to some probability distribution $w \in \Delta([0, 1])$, the effective discount factor at step $t$ is given by the distribution's raw moment $\Gamma^{(t)} = m_t$, where

$$m_t = \int_0^1 w(\gamma) \gamma^t d\gamma \tag{19}$$

Consider the Beta distribution. Its probability distribution function (Johnson et al., 1994) is given by the following expression:

$$f(x; \alpha, \beta) = \frac{1}{B(\alpha, \beta)} x^{\alpha-1} (1-x)^{\beta-1} \tag{20}$$

The raw moments can be obtained as follows:

$$
\begin{aligned}
\Gamma^{(t)} = m_t &= \int_0^1 x^t f(x; \alpha, \beta) \\
&= \int_0^1 x^t \frac{1}{B(\alpha, \beta)} x^{\alpha-1} (1-x)^{\beta-1} \\
&= \frac{1}{B(\alpha, \beta)} \int_0^1 x^{(\alpha+t)-1} (1-x)^{\beta-1} \\
&= \frac{1}{B(\alpha+\beta)} B(\alpha+t, \beta) \\
&= \frac{\Gamma(\alpha+\beta)}{\Gamma(\alpha)\Gamma(\beta)} \times \\
&\quad \times \frac{\Gamma(\alpha) \cdot \alpha \cdot (\alpha+1) \cdot \ldots \cdot (\alpha+t-1) \cdot \Gamma(\beta)}{\Gamma(\alpha+\beta) \cdot (\alpha+\beta) \cdot (\alpha+\beta+1) \cdot \ldots \cdot (\alpha+\beta+t-1)} \\
&= \frac{\alpha \cdot (\alpha+1) \cdot \ldots \cdot (\alpha+t-1)}{(\alpha+\beta) \cdot (\alpha+\beta+1) \cdot \ldots \cdot (\alpha+\beta+t-1)} \\
&= \prod_{k=0}^{t-1} \frac{\alpha+k}{\alpha+\beta+k} \tag{21}
\end{aligned}
$$

which proves Equation 17. We then consider the recurrence between consecutive $\Gamma^{(\cdot)}$ values

$$
\begin{aligned}
\Gamma^{(t+1)} &= \prod_{k=0}^{t} \frac{\alpha+k}{\alpha+\beta+k} \\
&= \frac{\alpha+t}{\alpha+\beta+t} \prod_{k=0}^{t-1} \frac{\alpha+k}{\alpha+\beta+k} \\
&= \frac{\alpha+t}{\alpha+\beta+t} \Gamma^{(t)} \tag{22}
\end{aligned}
$$

proving Equation 18 and completing our proof of Theorem 1.

$\square$

**Lemma 4.** *Special cases of Beta-weighted discounting*

*Consider a discounting $\Gamma^{(t)}$ given by the Beta-weighted discounting parametrized by $\mu \in (0,1), \eta \in (0,1]$. The following is true:*

- *if $\eta \to 0$, then $\Gamma^{(t)} = \mu^t$, i.e. it is equal to exponential discounting*
- *if $\eta = 1$, then $\Gamma^{(t)} = \frac{\mu}{\mu + (1-\mu)t} = \frac{1}{1+t/\alpha}$, i.e. it is equal to hyperbolic discounting*

*Proof.* Remember that $\mu = \frac{\alpha+\beta}{\beta}, \eta = \frac{1}{\beta}$. Let us first consider $\eta \to 0$, i.e. $\beta \to \infty$ so that $\frac{\alpha}{\alpha+\beta} = const$. Note that this also implies $\alpha \to \infty$. Consider the expression for $\Gamma^{(t)}$:

$$\Gamma^{(t)} = \prod_{k=0}^{t-1} \frac{\alpha + k}{\alpha + \beta + k} \tag{23}$$

As $\alpha$ and $\beta$ grow arbitrarily high, the bounded values of $k$ become negligible, and the expression can be reduced to

$$\Gamma^{(t)} = \prod_{k=0}^{t-1} \frac{\alpha}{\alpha + \beta} = \prod_{k=0}^{t-1} \mu = \mu^t. \tag{24}$$

For $\eta = 1$, we can reuse the expression obtained in the proof of Lemma 5. As shown there, with $\beta = \eta = 1$, the effective discount factor is

$$\Gamma^{(t)} = \frac{1}{1 + t/\alpha} \tag{25}$$

which with $k = \frac{1}{\alpha}$, becomes the usual hyperbolic discounting:

$$\Gamma^{(t)} = \frac{1}{1 + kt} \tag{26}$$

thus completing the proof.

$\square$

**Lemma 5.** *Beta-weighted discounting summability*

*Given the Beta-weighted discount vector $\Gamma^{(t)} = \prod_{k=0}^{t-1} \frac{\alpha+k}{\alpha+\beta+k}$, $\alpha \in [0, \infty)$, $\beta \in [0, \infty)$, the following property holds:*

$$\sum_{t=0}^{\infty} \Gamma^{(t)} = \begin{cases} \frac{\alpha+\beta-1}{\beta-1} & \text{if } \beta > 1 \\ \infty & \text{otherwise} \end{cases} \tag{27}$$

*Thus, Beta-weighted discounting is summable iff $\beta > 1$.*

*Proof.* First, we analyze the convergence of Beta-weighted $\Gamma^{(t)}$ depending on $\alpha, \beta$.

In particular, we consider the series:

$$S = \sum_{t=0}^{\infty} a_t = \sum_{t=0}^{\infty} \left( \prod_{k=0}^{t-1} \frac{\alpha + k}{\alpha + \beta + k} \right) \tag{28}$$

We then use the Raabe's convergence test (Ali, 2008). Given a series $(a_t)$ consider the series of terms $b_t = t \left( \frac{a_t}{a_{t+1}} - 1 \right)$ and its limit $L = \lim_{t \to \infty} b_t$. There are three possibilities:

- if $L > 1$, the original series converges
- if $L < 1$, the original series diverges

- if $L = 1$, the test is inconclusive

In the case of Beta-weighted discounting, we have:

$$
\begin{aligned}
\lim_{t \to \infty} b_t &= t \left( \frac{\prod_{k=0}^{t-1} \frac{\alpha+k}{\alpha+\beta+k}}{\prod_{k=0}^{t} \frac{\alpha+k}{\alpha+\beta+k}} - 1 \right) \\
&= \lim_{t \to \infty} \frac{t}{\frac{\alpha+t}{\alpha+\beta+t}} - t \\
&= \lim_{t \to \infty} \frac{\alpha t + \beta t + t^2 - \alpha t - t^2}{\alpha + t} \\
&= \lim_{t \to \infty} \frac{\beta t}{\alpha + t} \\
&= \beta
\end{aligned}
\tag{29}
$$

Thus, we show that Beta-weighted discounting is summable with $\beta > 1$ and nonsummable with $\beta < 1$. For $\beta = 1$, we can rewrite the effective discount factor as:

$$
\begin{aligned}
\Gamma^{(t)} &= \prod_{k=0}^{t-1} \frac{\alpha+k}{\alpha+\beta+k} \\
&= \prod_{k=0}^{t-1} \frac{\alpha+k}{\alpha+k+1} \\
&= \frac{\alpha \cdot \cancel{(\alpha+1)} \cdot \ldots \cdot \cancel{(\alpha+t-2)} \cdot \cancel{(\alpha+t-1)}}{\cancel{(\alpha+1)} \cdot \cancel{(\alpha+2)} \cdot \ldots \cdot \cancel{(\alpha+t-1)} \cdot (\alpha+t)} \\
&= \frac{\alpha}{\alpha + t} \\
&= \frac{1}{1 + t/\alpha}
\end{aligned}
\tag{30}
$$

The series $\sum_{t=0}^{\infty} \frac{1}{1+t/\alpha}$ is a general harmonic series and therefore divergent, completing the proof of convergence.

To obtain the exact value, we use the following Taylor expansion

$$
\frac{1}{1-x} = 1 + x + x^2 + \ldots
\tag{31}
$$

By evaluating the expected value of this expression with the Beta distribution's probability density function $w(x)$, we obtain the desired sum of all discount factors:

$$
\begin{aligned}
\mathbb{E}\left(\frac{1}{1-X}\right) &= \int_0^1 \frac{1}{1-x} f(x; \alpha, \beta) dx \\
&= \int_0^1 w(x) + x w(x) + x^2 w(x) + \ldots \, dx \\
&= \int_0^1 x^0 w(x) dx + \int_0^1 x^1 w(x) dx + \ldots \\
&= \Gamma^{(0)} + \Gamma^{(1)} + \ldots = \sum_{t=0}^{\infty} \Gamma^{(t)}
\end{aligned}
\tag{32}
$$

This expression can be expanded as follows:

$$
\begin{aligned}
\mathbb{E}(\frac{1}{1-X}) &= \int_0^1 \frac{1}{1-x} f(x; \alpha, \beta) dx \\
&= \int_0^1 (1-x)^{-1} \frac{1}{B(\alpha, \beta)} x^{\alpha-1} (1-x)^{\beta-1} \\
&= \frac{1}{B(\alpha, \beta)} \int_0^1 x^{\alpha-1} (1-x)^{(\beta-1)-1} \\
&= \frac{B(\alpha, \beta-1)}{B(\alpha, \beta)} \\
&= \frac{\Gamma(\alpha)\Gamma(\beta-1)}{\Gamma(\alpha+\beta-1)} \frac{\Gamma(\alpha+\beta)}{\Gamma(\alpha)\Gamma(\beta)} \\
&= \frac{\cancel{\Gamma(\alpha)}\cancel{\Gamma(\beta-1)}}{\cancel{\Gamma(\alpha+\beta-1)}} \frac{(\alpha+\beta-1)\cancel{\Gamma(\alpha+\beta-1)}}{(\beta-1)\cancel{\Gamma(\alpha)}\cancel{\Gamma(\beta-1)}} \\
&= \frac{\alpha+\beta-1}{\beta-1}
\end{aligned}
\tag{33}
$$

completing the proof.

$\square$

## B  Beta-weighted Discounting Properties

In Table 1 we present the values of the properties described in Section 5 for a set of discounting methods. For each of them, we list their normalized partial sums $\Gamma_0^{10}$, $\Gamma_{10}^{100}$, $\Gamma_{100}^{1000}$, $\Gamma_{1000}^{10000}$, the variance measure, the effective time horizon, and the total sum of the first 1000 steps.

Table 1: The values of different metrics for a chosen set of discounting method and their parameters.

| Discounting method | $\Gamma_0^{10}$ | $\Gamma_{10}^{100}$ | $\Gamma_{100}^{1000}$ | $\Gamma_{1000}^{10000}$ | Variance $\sum_{t=0}^{10000} \Gamma^{(t)2}$ | $T_{eff}$ | Total $\sum_{t=0}^{1000} \Gamma^{(t)}$ |
|---|---|---|---|---|---|---|---|
| No discounting | 0.001 | 0.009 | 0.090 | 0.900 | 10000 | 6322 | 1000 |
| Exponential $\gamma = 0.99$ | 0.096 | 0.538 | 0.366 | 0.000 | 50.25 | 100 | 100 |
| Exponential $\gamma = 0.999$ | 0.010 | 0.085 | 0.537 | 0.368 | 500.25 | 1000 | 632.3 |
| Exponential $\gamma = 0.97$ | 0.263 | 0.690 | 0.0480 | 0.000 | 16.92 | 33 | 33.3 |
| Beta-weighted $\mu = 0.99$, $\eta = 0.5$ | 0.049 | 0.293 | 0.509 | 0.149 | 66.67 | 323 | 166.1 |
| Beta-weighted $\mu = 0.97$, $\eta = 0.5$ | 0.135 | 0.476 | 0.334 | 0.055 | 22.23 | 110 | 61.7 |
| Hyperbolic $\mu = 0.99$ | 0.021 | 0.130 | 0.370 | 0.479 | 98.53 | 1741 | 238.8 |
| Hyperbolic $\mu = 0.25$ | 0.439 | 0.188 | 0.187 | 0.187 | 1.12 | 107 | 3.3 |
| Fixed-horizon $T_{max} = 100$ | 0.100 | 0.900 | 0.000 | 0.000 | 100 | 64 | 100 |
| Fixed-horizon $T_{max} = 160$ | 0.062 | 0.562 | 0.375 | 0.000 | 160 | 102 | 160 |
| Truncated Exponential $\gamma = 0.99$, $T_{max} = 100$ | 0.151 | 0.849 | 0.000 | 0.000 | 43.52 | 51 | 63.4 |
| Truncated Exponential $\gamma = 0.99$, $T_{max} = 500$ | 0.096 | 0.542 | 0.362 | 0.000 | 50.25 | 99 | 99.3 |
| Truncated Beta-weighted $\mu = 0.99$, $\eta = 0.5$, $T_{max} = 100$ | 0.143 | 0.857 | 0.000 | 0.000 | 47.11 | 54 | 69.4 |
| Truncated Hyperbolic $\mu = 0.99$, $T_{max} = 100$ | 0.138 | 0.862 | 0.000 | 0.000 | 50.13 | 55 | 69.4 |
| Truncated Hyperbolic $\mu = 0.99$, $T_{max} = 500$ | 0.054 | 0.335 | 0.612 | 0.000 | 83.13 | 210 | 178.6 |

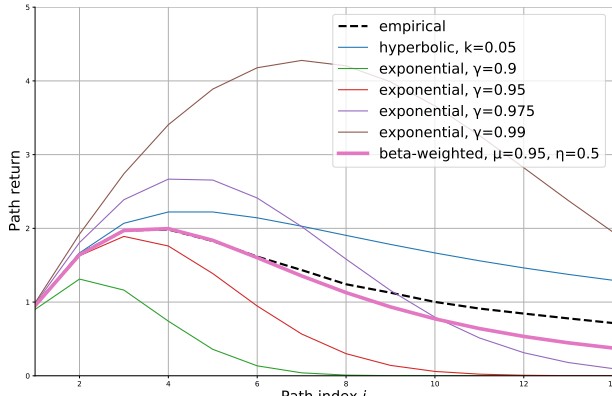

Figure 5: Pathworld environment results under different discounting schemes. Hyperbolic (Fedus et al., 2019) and exponential (various curves) discountings fail to approximate the empirical (dashed) value. Instead, the proposed Beta-weighted discounting approximates it much better, despite its different functional form.

## C   Pathworld experiments

Table 2: Values of the Mean Square Error for different discounting methods on the Pathworld environment, summed across the first 14 paths $i \in \overline{1, 14}$. Lower is better.

| DISCOUNTING METHOD | DISCOUNT FACTOR† | $\eta$ | MSE |
|---|---|---|---|
| EXPONENTIAL | 0.990 | 0 | 3.962 |
| EXPONENTIAL | 0.950 | 0 | 0.446 |
| EXPONENTIAL | 0.975 | 0 | 0.242 |
| HYPERBOLIC‡ | 0.05 | 1 | 0.250 |
| BETA-WEIGHTED | 0.95 | 0.5 | **0.032** |

†: FACTOR IS $\gamma$ FOR EXPONENTIAL
   $k$ FOR HYPERBOLIC
   $\mu$ FOR BETA-WEIGHTED
‡: RESULTS OBTAINED BY OUR RE-IMPLEMENTATION
   OF FEDUS ET AL. (2019)

We showcase the utility of our method on a simple toy environment called the Pathworld introduced by Fedus et al. (2019). Our goal is to show that Beta-weighted discounting can accurately model the presence of unknown risk in an environment, even without being designed to a priori match the functional form of the risk distribution.

### C.1   Setup

In Pathworld, the agent takes a single action and observes a single reward after some delay. The actions (i.e. paths) are indexed by natural numbers. When taking the $i$-th path, the agent receives a reward $r = i$ after $d = i^2$ steps. Each path may also be subject to some hazard. In each episode, a risk $\lambda$ is sampled from the uniform distribution $U([0, 2k])$ for a given parameter $k$. Given the risk $\lambda$, at each timestep on the path, the agent has a chance $(1 - e^{-\lambda})$ of dying, and thus not collecting any reward in that episode. The task of the agent is to use experience gathered without risk, and use the discounting to accurately predict a path's value when evaluated with an unknown risk.

## C.2 Pathworld results

If the risk is sampled from the Dirac Delta distribution $\mathcal{H} = \delta(\lambda - \lambda_0)$, the optimal discounting method is exponential discounting, i.e. $\Gamma^t = \gamma^t$ with $\gamma = e^{-\lambda}$. As shown by Fedus et al. (2019), if the risk is sampled from an exponential distribution $\mathcal{H} = \frac{1}{k}\exp(-\lambda/k)$, the optimal discounting scheme is their hyperbolic discounting $\Gamma^t = \frac{1}{1+kt}$.

As shown in Lemma 4, Beta-weighted discounting subsumes both exponential ($\mu{=}\gamma$, $\eta{\to}0$) and hyperbolic discounting ($\alpha{=}\frac{1}{k}$, $\eta{=}1$). Thus, it directly models scenarios whose optimal discounting is exponential or hyperbolic.

A more interesting scenario is when the functional form is different, e.g. when the risk is sampled from a uniform distribution $\mathcal{H} \sim U([0, 2\lambda_\mu])$. Figure 5 and Table 2 report the results. We observe that Beta-weighted discounting (with $\mu$ chosen to fit the mean of the true risk distribution, and $\eta$ chosen heuristically to decrease the variance of the Beta distribution) successfully outperforms all baselines, indicating that using Beta-weighted discounting enabled by the proposed UGAE allows better modelling of unknown risk distributions in environments where the risk phenomenon makes discounting necessary.

