# OpenReview forum: "UGAE: A Novel Approach to Non-exponential Discounting"
_TMLR — Rejected by TMLR_

### Review · Reviewer_xAyN · 2023-04-13

**Summary Of Contributions:**

1. The authors extend GAE to work with any choice of discounting (UGAE).

2. The authors generalize exponential and hyperbolic discounting by proposing a beta-weighted discounting.

3. The authors claim that UGAE can reach better performance empirically.

**Audience:**

No

**Claims And Evidence:**

No

**Requested Changes:**

*Critical changes*

(1) Figure 1 and all relevant text (section 5 – 5.4) are extremetly unclear to me. I don’t understand what the authors are trying to show:

a. The legend is unclear – what is the meaning of every color?

b. Is there an environment with real rewards, or is this just a demonstration that depends only on the IID normally distributed rewards and constant choices of discounting?

I believe that if the legend would have made sense to me the rest of the understanding would follow.

(2) The authors compare empirically between MC and UGAE, but not between UGAE and GAE. Why? If UGAE is under-performing, perhaps there is not much to gain from tampering with the discount factor. Also, the experiment setup seems not enough for me – since adapting the code is relatively simple – I think many more experiments could have been relatively easily obtained with one of the commonly used packages.

*Strengthen the work*

(1) "In practice, γ is one of the most important hyper-parameters to tune. Small deviations may massively decrease the algorithm’s performance, which makes training an RL agent less robust, and more difficult for users to perform a hyper-parameter search with new algorithms and environments."

I actually am not familiar with such strong claim about the discount factor, and as far as I remember many empirical papers do not sweep over the discount factor. I would ask the authors to either add a reference, or tone down the claim.

(2) "In particular, δ V t cannot be used as its value depends on which timestep’s advantage we are computing."

I did not understand this sentence, and would ask the authors to make it clearer.

(3) "… due to the need for multiplying large vectors."
Please make it a bit clearer that now in every step there is vector multiplication.

**Strengths And Weaknesses:**

*Strengths*

(1) The GAE formulation is commonly used, which means that any intuition and changes can have large impact and will interest the community.

(2) The proposed changes are clean and simple to embed in any implementation.

(3) Most of the paper is clearly written, I went over the proofs and they seem correct to me.

*Weaknesses*

(1) I am far from being conceived that tampering with the discount factor has any justification. As the authors mentioned, their method do have some (albeit not high) computational cost and technical cost. But I saw no upsides to doing so (theoretical or empirical).
The experimental part of the paper – the paper considers two experimental setups.

(2) The first is for showing the properties of different discount choices. While its currently not entirely clear to me (see requested changes), I believe the results there would make sense and probably immediately follow the simple mathematical formulation beyond each discounting choice. Note that these intuitions and conclusions are only interesting if I can deduce something practical from them, but I don’t see what.

(3) The second part, should convince the reader there is any advantage in changing the discount factor, but the comparison is far from showing that is the case.

Without any theoretical and practical justification for UGAE and beta-weighted discounting (that require more code, run time and parameter tuning), the paper is very unlikely to interest researchers from the field.

---

> ### Author Response · Authors · 2023-04-30
>
> Thank you for the review, we will apply all mentioned changes to the revised version. To address the specific questions:
>
> *Weaknesses*
> 1. Our justification for tampering with the discounting is predominantly to enable a greater flexibility of behaviors that can be obtained via RL. The main example is humans -- we tend to behave with non-exponential discounting, so it is important to keep in mind if we try to create a human-like agent (as opposed to a simple reward-maximizing agent)
> 2. See below
> 3. See below
>
> *Critical changes*
> 1. a. The colors simply refer to the different lines present in the graph. Each line represents one specific discounting, parametrized by the discount factor and truncation time horizon. For example, the blue and red lines differ by the discount factor going from $0.9$ to $0.999$.  The green and pink lines differ by introducing a truncation at $500$ steps.
>
> 1. b. This analysis is independent of an environment, and focuses only on the discounting. Only the variance estimation uses a "fake" environment, for which the assumptions are stated in Section 5.1
>
> 2. The goal of the empirical evaluation is showing that UGAE mirrors the performance benefits of GAE, but can be applied to non-exponential discounting methods. It is not to show that UGAE will outperform GAE on the general task of basic the undiscounted reward -- we do not expect this to be true in general. While we could run additional experiments, we do not believe this to be necessary just for the sake of making the number larger -- the original GAE paper similarly used 4 environments, and has since been accepted as the de-facto standard advantage estimation method for exponential discounting.
>
>
> *Strengthen the work*
>
> 1. A good description of this phenomenon is in the blog post by C. Tessler: https://tesslerc.github.io/posts/drl_works_now_what/
> The author shows there an example of the Hopper-v2 performance, which shows a range of $\gamma \in [0.985, 0.995]$ in which the training performs well, but outside of that range, the reward drops significantly.
>
> 2. This refers to the fact that only the exponential discounting has the property of $\Gamma^{(t_1)} * \Gamma^{(t_2)} = \Gamma^{(t_1 + t_2)}$. This means that when processing the reward obtained in step $50$, it makes no difference it is for the advantage at step $0$, $1$ or $30$. This allows for sharing parts of the computation when estimating the advantages in different steps. On the other hand, this property is not preserved with non-exponential discounting, which makes sharing this computation impossible, and thus when we process the reward from step $50$, we have to process it separately for the advantage at step $0$, $1$ and so on.
>
> 3. We will rephrase this for better clarity

---

> > ### Comment · Reviewer_xAyN · 2023-05-02
> > **I have read your reply**
> >
> > Dear authors, thank you for your reply.
> >
> > - Thank you for clarifying Figure 1 for me.
> >
> > - Re strengthen the work 1: Thanks for the reference, I urge you to add the paper discussed in the blog to support your claim.
> >
> > - Re strengthen the work 2: Thanks for clarifying this point.
> >
> > Despite your answer to critical changes 3 (motivation - a point which was also raised by all other reviewers), I think that showing the proposed UGAE does behaves similarly to GAE is not enough, and you need to show its benefit.
> >
> > Exponential discounting is simpler than UGAE and beta-weighting, it is easy to compute and holds very nice theoretical properties like being memoryless and giving closed form analytical expressions for many results. In my view, the bar for researchers to take interest in this work is higher since you propose something new that does not hold all of these properties. It needs to surpass exponential discounting in something -- higher rewards in specific environments, better mimicking human behavior empirically (maybe imitation learning regime?), better bias-variance trade-off (as proposed by one of the other reviewers) or any other regime, to justify its usage - otherwise (again in my perspective) the work is not full.

---

### Review · Reviewer_ZUVR · 2023-04-17

**Summary Of Contributions:**

The paper addresses the problem of general (non-exponential) discounting in reinforcement learning (RL). The paper introduces the notion of universal generalized advantage estimation (UGAE) that extends GAE for general discounting function, discussing the computational challenges of the approach. Then, a specific class of discounting is considered, i.e., the one in which a combination of exponential discounting with different values of the discount factor is performed. The weight assigned to each of the different values of the discount factor is selected using a Beta distribution. Specific particular cases are investigated. A numerical validation is finally reported to highlight the behavior of the proposed apporach.

**Audience:**

Yes

**Broader Impact Concerns:**

None.

**Claims And Evidence:**

No

**Requested Changes:**

Please address the concerns raised in the Weaknesses section. In particular:
- Clarify the contribution of the paper and discuss its significance.
- Clarify in a quantitative manner the effect of truncating the trajectories.
- Explain and justify the choices made in the experimental evaluation.

**Strengths And Weaknesses:**

**Strengths**
- The paper is well-written.
- The paper has the merit of studying a generalization of the exponential discounting, which might be of interest for the community.

**Weaknesses**
- [Contribution] It is not fully clear which is the main contribution of the paper. From the title and the introduction, it seems that the main contribution is UGAE, although a large portion of the paper is devoted to the discussion of the Beta-weighted discounting. Furthermore, I missed the connection between the two parts. Is the Beta-weighted discounting particularly suitable for UGAE, compared to other kinds of discounting? It is just a choice?
- [Definition of UGAE] The proposed UGAE combines the n-step returns using the standard exponential mean based on the parameter $\lambda$. This is fully natural when considering exponential discounting. I am wondering whether in presence of a different form of discounting non-exponential means would perform better.
- [Computational issues and bias] In Section 3.1, the authors discuss the computational properties of the UGAE estimation that lead to more expensive computations compared to standard GAE. Furthermore, the authors propose to truncate the trajectories to a maximum length to control this computational burthen. However, it is not clear how this truncating affects the bias in a quantitative manner. Theorem 2 does not properly illustrate this point, unfortunately. Can the authors elaborate on this point?
- [Assumptions] The assumption in paragraph "variance of the discounted rewards" that the "rewards to be uncorrelated" seems quite restrictive.
- [Beta-weighted discounting] The paper lacks a theoretical analysis of the bias-variance trade off (mentioned by the authors in Section 5.3) of the Beta-weighted discounting as a function of the parameters of the Beta distribution. This would provide a justification (if any) of the use of such an approach for discounting.
- [Experiments] The experimental evaluation is not fully convincing for several reasons. First, the choice of the $\lambda$ parameter values seems arbitrary. Is any hyperparameter optimization approach run for this purpose? Not all the environments employ the same set of $\lambda$ values. Second, a large discussion about the use of $\eta \in \{0,1\}$ is provided earlier in the paper (in relation with exponential and hyperbolic discounting), but these values are not tested empirically.

---

> ### Author Response · Authors · 2023-04-30
>
> Thank you for your review. Addressing the comments in the weaknesses:
>
> 1. [Contribution] Our paper introduces both UGAE and Beta-weighted discounting. These two concepts can, in principle, exist separately, but in practice, each of them would be impractical without the other one. Just introducing Beta-weighted discounting would not be very impactful without a way of using it in practice (e.g. UGAE). At the same time, it would be difficult to use UGAE without non-exponential discountings that it could implement. Beta-weighted discounting provides a convenient parametrization for a family of non-exponential discountings that can be used with UGAE.
> 2. [Definition of UGAE] It is possible that a non-exponential weighing would have interesting properties, but we do not believe that this is necessary for non-exponential discounting -- $\gamma$ has a clear temporal definition, and while $\lambda$ is also connected to the temporal aspect, all the estimators $\hat{A}^{(k)}$ fundamentally compute the same value (but with a different bias-variance tradeoff)
> 3. [Computational issues and bias] The effect of the truncation is fairly simple, in that the agent does not learn from rewards which are too distant in time from the given action. Our main point here is that this is a method for decreasing the computational complexity with very long episodes, while maintaining some properties of the arbitrary non-exponential discounting. In a vast majority of practical RL problems, this is not a significant concern.
> 4. [Assumptions] While it is restrictive, this analysis is meant to abstract away the environment and the policy, and give an idea of the scale scale of the variance of the advantage estimation. Correlations between the rewards would make the analysis dependent on how exactly they are correlated, so we use this assumption to make the analysis tractable.
> 5. [Beta-weighted discounting] With the mention of the bias-variance trade-off, we mainly refer to the general presence of this phenomenon in temporal discounting. Even without any non-exponential discounting, a lower discount factor is meant to decrease the variance, but necessarily increases the bias. We wanted to bring attention to this point, as it is a reocurring theme in the paper. Additionally, the main rationale for using non-exponential discounting is for situations where researchers or practitioners want to emulate specific types of agents, e.g. humans, who tend to behave non-exponentially.
> 6. [Experiments] The main goal of our experiments is showing that using UGAE mirrors the performance benefits of GAE, but with non-exponential discounting. For this reason, we use values of $\lambda$ which are known to perform well with GAE from prior work. As for $\eta$, its purpose is parametrizing the discounting mechanism, so we show that for various values of $\eta$, using UGAE is beneficial over MC. We will be happy to run additional experiments with different values of $\eta$ for the revised version of this paper, but from our current experiments (which did not make their way into the paper) it seems clear that they will be roughly identical to the figues currently present in the paper.

---

### Review · Reviewer_ib4D · 2023-04-20

**Summary Of Contributions:**

This paper offers two main contributions:
1. It proposes the Universal Generalized Advantage Estimator (UGAE), which is a generalization of GAE [Schulman et al., 2018] and allows for computation of GAE advantages with arbitrary discounting.
2. It proposes Beta-weighted discounting, which is a generalization of both exponential and hyperbolic discounting, which lie at the extremes of a spectrum Beta-weighting enables.

The authors provide some empirical analyses of their proposals to justify their use.

**Audience:**

Yes

**Broader Impact Concerns:**

No major concerns.

**Claims And Evidence:**

Yes

**Requested Changes:**

I would suggest better motivating the work, in a way that more clearly ties together the beta-weighting proposal and UGAE. The empirical evaluations need to be strengthened, as right now it's simply suggesting that UGAE _could_ lead to improvements, but it's not at all clear how future work could leverage this in other environments.

Some more detailed suggestions and questions for improvement:
1. In the **Problem Setting** section, you're using $r_t$ both as a random variable ($r_t = R(s_t, a_t)$) and a real number (in trajectory). This causes confusion further down (see note below).
1. In section 3 it would be good if you clarified more (beyond brief discussion in Section 2) how GAE is used for control.
1. From section 3.1 onwards, it seems like you're assuming $\pi$ is fixed? Please be explicit about this.
1. When defining $A^{GAE}_t$, what does it mean if $\lambda = 1$? It would be useful to discuss this, as it is mentioned in the DRL experiment section.
1. In Theorem 2, what do you mean by "an arbitrary summable discounting $\Gamma^{(t)}$?
1. I'm confused by equation (5). I thought the summation was over $t$, but you have $dt$, suggesting the _integral_ is over $t$. Which is it?
1. It's not clear where the sentence immediately following equation (5) ("An important observation...") comes from. Why does this hold? A brief discussion would be useful.
1. In the sentence right above section 5.1, what do you mean by "We focus on a characteristic time scale of the environment around 100 steps."?
1. In the first paragraph of section 5.1 you say "the longer the considered time horizon is, the more timesteps it includes, increasing its overall importance", but it's not clear what "its" is referring to.
1. The **Importance of future rewards** paragraph should be rewritten, as it's hard to grok at the moment. It's not clear what you're trying to convey.
1. In equation (9), what are $t_1$ and $t_2$?
1. Equation (10) is missing the "(10)" to its right
1. Immediately above equation (10), are you assuming the same $\mu$ for all $t$? Independent of state and policy? This seems like an unreasonably strong assumption.
1. It's not clear how the **Total sum of the discounting** paragraph is a "Property of discounting", as suggested by the section title. It seems more like you're explaining design choices for your evaluations.
1. Does the discussion in section 5.4 suggest hyperbolic discounting places more importance on future rewards than exponential discounting, as per Lemma 4?
1. The discussion in the second paragraph of section 5.5 (on "inconsistent behavior") is a little hand-wavy and merits better formalization.
1. In Section 6, how are the Monte Carlo advantage estimates computed in practice?
1. In page 10 you say "relatively far from $\lambda = 1$, based on prior work". What prior work? Citations?
1. You say"A single training takes 2-5 hours with a consumer GPU.". Is this for all environments?

Minor fixes:
1. In second paragraph of page 3, should be "TRPO **and** PPO".
1. In the intro paragraph of section 4, no need for the "i.e." before "Beta-weighted"

**Strengths And Weaknesses:**

# Strengths
This paper proposes a nice generalization of both GAE and traditional discounting methods via their Beta-weighting. The theoretical results* demonstrate that Beta-weightings are well-formed and well-behaved. This generalized way of expressing discounting factors can prove quite useful to the RL community.

# Weaknesses
I have two main concerns with this paper:
1. **Cohesion** Although it seems like beta-weightings are needed for UGAE formulation, the paper currently reads as two separate ideas stitched together. Section 3 deals with UGAE, sections 4 and 5 deal with beta-weightings, and then section 6 goes back to just UGAE. I would have expected to see more discussion of beta-weightings (and different value choices) in the experiments in section 6. It's not even clear what type of discounting was used for them.
2. **Motivation** The motivation is not entirely clear. While interesting, it's not clear why one would use the proposed methods. See also point below.
3. **Empirical evaluation** I don't find the empirical results convincing enough, nor making a strong enough claim for the paper's proposals. Specifically:

     a. It's difficult to know what to make of Figure 1. The discussion in section 5.3 talks about "no discounting", "exponential discounting", "fixed-horizon discounting", etc., but it's not clear which of the lines in Figure 1 they are referring to. Given that there are quite a few hparams varying, a little more hand-holding for the reader would be useful.

     b. It's not clear how the findings from section 5 apply to the experiments in section 6. Did they inform the tuning at all? How so?

     c. In Figure 3 both $\lambda$ and $\eta$ were tuned for each for each separate environment, which is non-standard when evaluating new methods. How were these tuned? Why are they only compared against MC (which did not receive any tuning)? How would one choose the various hparams for beta-weightings when tackling a new problem?

     d. Some specific design choices were made (e.g. episode length of 10,000, "charateristic time scale" of 100, choice of 37% as effective horizon), and it's not clear why they were made nor how sensitive evaluations are to these choices.


[*] Note that I did not yet review the proofs, as I wanted to enter the review, as I was already late in doing so. Will go through the proofs after submitting this.

---

> ### Author Response · Authors · 2023-04-30
>
> Thank you for your review. To address the specific comments:
>
> Weakness 1: While there are indeed two ideas which could, in principle, be separate, we decided to introduce both of them in this paper for practical reasons. With the currently existing methods, using Beta-weighted discounting makes little sense without an equivalent to UGAE. At the same time, UGAE is difficult to apply without a well-parametrized non-exponential discounting. So in a way, both of these methods rely on one another to be applicable.
> In section 6 we use Beta-weighted discounting as we state in the first paragraph of that section, but we will make this clearer in the revision.
>
> Weakness 2: We intend for our work to enable further empirical research into non-exponential discounting methods. Our main argument for why this is relevant, is that humans themselves tend to act with a non-exponential discounting, so it is important to keep this in mind when designing human-like agents. We expand on this in Section 5.5
>
> Weakness 3: Our experiments are meant to show that the performance benefits provided by UGAE over non-exponential Monte Carlo, mirror the benefits of GAE over regular exponential Monte Carlo.
> Sections 5 and 6 are largely meant to address different aspects of understanding our new approach. Section 5 analyzes the properties of various discounting methods, showing that they are rather diverse, and thus it is relevant in certain cases to use non-exponential methods.
> Section 6 is a simple empirical evaluation of the aforementioned performance benefit.
>
> In Figure 3, we took an approach similar to the original GAE paper. We do not exactly tune the $\eta$ values, as they serve the role of parametrizing the non-exponential discounting. The $\lambda$ values are indeed chosen for each environment separately, according to prior work. Doing otherwise would result in a risk of unreliable results due to suboptimal hyperparameter choices.
>
> When tackling a new problem, the $\eta$ parameter would depend on the problem at hand -- simulating a human could imply $\eta = 1$
> for hyperbolic discounting, following some prior work. If summability of the discounting is important for a continuing problem, then a slightly smaller value like $\eta = 0.9$ could be more appropriate, to maintain the inconsistency of the discounting, but keep it summable.
> The choice of $\lambda$ can be done in the exact same way as with GAE.
>
> As for the design choices, the time scale of 100 is simply meant to reflect how long episodes are in commonly encountered RL environments, precise to the order of magnitude. The episode length of 10,000 is meant to be significantly greater than that to emulate "infinity", but keep things computable. The choice of $\frac{1}{e}=0.37 $ as the time horizon is based on the way to define a time horizon from prior work, e.g. from "Near-Optimal Reinforcement Learning in Polynomial Time" (Kearns and Singh, 2002)
>
>
> Requested changes (we will address all of them in the revision, and answer the questions here):
>
> 5. We mean any discounting function $\Gamma^{(\cdot)}$ as defined in Section 3, such that $\Sigma_0^\infty \Gamma^{(t)} < \infty$
> 6. This is a typo, the integral was meant to be $d\gamma$, thank you
> 7. This comes from the definition of distribution moments, we will make it more clear
> 8. We assume that most relevant consequences in the environment happen within around 100 steps, which generally implies a discount factor of around 0.99. If we were dealing with much longer episodes (e.g. timestep of an underlying simulation scaled down by a factor of 10), then a starting point of an exponential discount factor would be closer to 0.999
> 11. Abstractly, $t_1, t_2$ are just arguments to the new introduced symbol $\Gamma_{t_1}^{t_2}$. Practically, in that part we consider the ratio of (cumulative reward between $t_1$ and $t_2$) and (cumulative reward between $0$ and $t_1$)
> 13. While it is a strong assumption, our analysis is aimed at the properties of the discounting, so we have to abstract away the policy and the environment to arrive at a meaningful interpretation.
> 14. The main reason for including the total sum is its finiteness, which is a very important property of a discounting. In practice, if we want to use a different discounting whose total sum is much higher than that of our previous experiments, other parts of the algorithm might have to be adjusted.
> 15. Hyperbolic discounting puts a higher weight on the very distant future as compared to exponential discounting, because it does not decay as quickly as the exponent.
> 17. In practice, they are computed using the same function as the UGAE implementation, which is mathematically equivalent to computing MC
> 18. We base it on the sources from which we took the respective parameters - (Raffin, 2020) and (Kwiatkowski et al. 2023)
> 19. Yes, for all environments we used in the experiments. The exact runtime depends on many factors, but this is meant to provide a basic intuition.

---

### Decision · Action_Editors · 2023-05-22

**Recommendation:** Reject

**Comment:**

This article studies non-exponential discounting in reinforcement learning through two lens: an extension of GAE, termed Universal GAE, providing a (biased) estimate of the value function for general discounting, and beta-weighted discounting, an interpolation between exponential and discounted discounting retrieved as special cases. The article provides some numerical illustrations of beta-discounting and an experiment comparing UGAE to MC rollouts.

The reviewers are unanimous in stating that the paper is not ready for publications. The two contributions (UGAE and beta-weighting) are related, but could be presented in a more cohesive way. The paper does not provide strong enough motivations for the proposed contributions. It also does not provide enough theoretical and/or practical evidences.

Below are some suggestions that could strengthen the paper:
* with non-exponential discounting, many facts about MDPs do not hold. For example, the optimal policy is likely to be time-dependent (only stationary policies are considered), bootstrapping no longer holds (while it is used in UGAE; the induced bias is shortly discuss, but the result, that the bias is finite, is not very informative, as a zero estimator would also enjoy this property), policy improvement or policy gradient no longer hold, at least in their classic form (while PPO is used here, without mentioning this aspect), etc. All these aspects should be discussed, and if some of them are heuristically considered, some related justifications or empirical evidences should be provided
* the empirical evidences should go beyond a comparison between UGAE and MC rollouts. They should showcase settings where non-exponential discounting is the thing to do, compare it to exponential discounting, justify the use of the modified value function within what is essentially a policy gradient relying strongly on exponential discounting, and discuss what kind of beta-discounting to choose and when (the classic discount parameter can be seen as a hyperparameter, but here there are more, and less well understood)


**Audience:**

Yes

**Claims And Evidence:**

No